# PAPERCLIP: Associating Astronomical Observations and Natural Language with Multi-Modal Models

**Siddharth Mishra-Sharma, Yiding Song, & Jesse Thaler**
MIT & IAIFI
{smsharma,ydsong,jthaler}@mit.edu

## Abstract

We present PAPERCLIP (Proposal Abstracts Provide an Effective Representation for Contrastive Language-Image Pre-training), a method which associates astronomical observations imaged by telescopes with natural language using a neural network model. The model is fine-tuned from a pre-trained Contrastive Language–Image Pre-training (CLIP) model using successful observing proposal abstracts and corresponding downstream observations, with the abstracts optionally summarized via guided generation using large language models (LLMs). Using observations from the *Hubble* Space Telescope (HST) as an example, we show that the fine-tuned model embodies a meaningful joint representation between observations and natural language through quantitative evaluation as well as tests targeting image retrieval (i.e., finding the most relevant observations using natural language queries). and description retrieval (i.e., querying for astrophysical object classes and use cases most relevant to a given observation). Our study demonstrates the potential for using generalist foundation models rather than task-specific models for interacting with astronomical data by leveraging text as an interface.

## 1 Introduction

Machine learning (ML) is starting to have a significant impact in the sciences, with astrophysics being no exception. ML methods have demonstrated promise at every stage of the research pipeline, from instrument design, to data acquisition, to its analysis (Huertas-Company & Lanusse, 2022). Most applications of ML within astrophysics have focused on augmenting traditional techniques in order to improve performance on specific tasks. The foundation model paradigm, in contrast, seeks to develop generalist models which can be deployed to simultaneously tackle a wide range of tasks (Bommasani et al., 2021). These models are typically pre-trained on massive amounts of unlabeled data using self-supervised or weakly-supervised learning techniques, enabling them to learn powerful representations which can then be used downstream. Foundation models can often benefit from additional training (fine-tuning) using a relatively small amount of domain-specific data in order to increase their usefulness when applied to specialized domains.

There is considerable interest in developing custom foundation models for the sciences (e.g., Batatia et al., 2023; Subramanian et al., 2023; McCabe et al., 2023; Birk et al., 2024; Vig et al., 2024; Heinrich et al., 2024), with astrophysics being ripe for such an effort given the large amounts of publicly-available data and diverse ways of interacting with it. The multi-modality inherent to astrophysical observations, with different types of data (e.g., images, spectra, light curves, textual descriptions) often available for a given target object, presents a unique opportunity.

In this paper, we describe PAPERCLIP (Proposal Abstracts Provide an Effective Representation for Contrastive Language-Image Pre-training), a method that connects, for the first time, astronomical image observations with natural language by leveraging the association between abstracts of successful observing proposals written by astronomers and images corresponding to downstream observations imaged by telescopes. This approach demon-

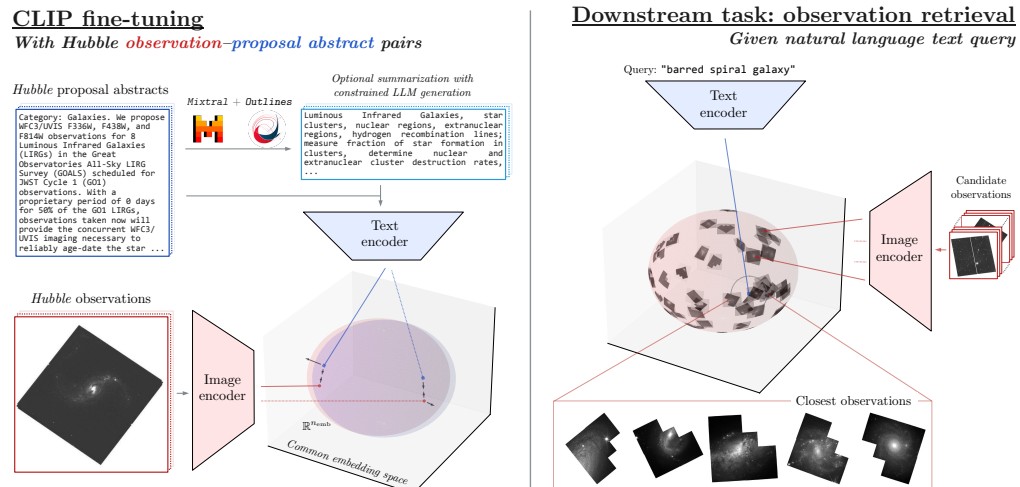

Figure 1: Overview of the PAPERCLIP method. (Left) A pre-trained CLIP model is fine-tuned using a dataset of *Hubble* observations and corresponding proposal abstracts. The proposal abstracts are optionally summarized using guided large language model generation. (Right) The fine-tuned model can then be used for downstream tasks such as observation retrieval i.e., finding the observations most relevant to a given text query. The proposal abstract snippet shown here corresponds to proposal ID 16914.

strates the potential of adapting generalist multi-modal foundation models to astronomy, complementing task-specific models by providing a flexible, language-based interface for interacting with observational data. Concretely, we showcase the method using observations imaged by the *Hubble* Space Telescope (HST). We show that fine-tuning a pre-trained CLIP (Contrastive Language-Image Pre-training; Radford et al., 2021) image-text model on observation-abstract pairs results in meaningful joint representations through quantitative and qualitative evaluation tests. Our method opens up the possibility of interacting with astronomical survey data using free-form natural language as an interface, which is a cornerstone of the success of the modern foundation model paradigm. A high-level overview of the method is shown in Fig. 1.

**Related Work** The concept of learning task-agnostic representations via self-supervised and contrastive learning has been applied within astrophysics (Slijepcevic et al., 2024; Stein et al., 2021; Hayat et al., 2021b; Slijepcevic et al., 2022) and used for downstream tasks like object similarity search (Stein et al., 2021), gravitational lens finding (Stein et al., 2022), estimation of Galactic distances (Hayat et al., 2021a), identification of rare galaxies (Walmsley & Scaife, 2023), and data compression (Akhmetzhanova et al., 2024). For a recent review of contrastive learning in astrophysics, see Huertas-Company et al. (2023). Beyond applications to a single modality, ASTROCLIP (Lanusse et al., 2023) recently used contrastive learning to learn a joint representation between galaxy images and associated spectra, showing that the learned representation embodies relevant physical properties and can be effectively used for downstream tasks like redshift and mass estimation. Bowles et al. (2023; 2022) introduced a method to associate radio galaxy images with a natural language description of their morphology by using human-generated descriptions, with the goal of deriving semantic morphology classes and using them for classification. In contrast with previous work, our application is the first to associate astronomical observation with the text modality in a task-agnostic manner, showcasing the potential of language models in specialized scientific domains like astronomy.

The rest of this paper is organized as follows. In Sec. 2, we describe the *Hubble* dataset used in this work, including the curation and processing of observations as well as text captions. In Sec. 3, we describe the methodology used to train and evaluate the model. In Sec. 4,

we present quantitative and qualitative results of our experiments on retrieval tasks. We discuss future prospects and conclude in Sec. 5.

## 2  Dataset Construction

We curate a dataset of *Hubble* Space Telescope (HST) image observations and corresponding text descriptions from publicly available sources. We rely on proposal abstracts from the Proposal Abstracts Catalog[1] – a catalog of successful HST proposals – to generate captions for the observations, optionally summarizing them via guided generation using LLMs (described in Sec. 2.2 below). The HST has been operational since its launch on April 24, 1990, and we use available proposals and observations up to the Cycle 30 science program, which commenced data-taking in 2022.

Table 1 shows examples of images and their corresponding (clipped) proposal abstracts. It can be seen that the images in this dataset exhibit specific characteristics as well as artifacts particular to HST data-taking and processing which distinguishes them from the distribution of natural images typically used for large-scale pre-training of foundation models. This further motivates fine-tuning on domain-specific data.

| *Hubble* image | Obs. cycle (Year) | Prop. ID | Proposal abstract (clipped) |
|---|---|---|---|
|  | 7 (1999) | 7340 | Category: STELLAR EJECTA. We propose to use the WFPC2 and STIS CCD to obtain maximum spatial resolution emission-line images of the young, oxygen- rich supernova remnants SN0540–69.3 in the LMC and E0102.2– 7219 in the SMC. O IIILambda5007, S IILambdaLambda6724 and O IILambdaLambda3727 images of SN0540–69.3 will be used to characterize the ionization structure and... |
|  | 22 (2016) | 13757 | Category: HOT STARS. Type Ia supernovae (SN Ia) have enormous importance to cosmology and astrophysics, but their progenitors and explosion mechanisms are not known in detail. Recently, observations and theoretical models have suggested that not all thermonuclear white-dwarf supernova explosions are normal SN Ia. In particular, type Iax supernovae (peculiar cousins to SN Ia), are... |

Table 1: Examples of *Hubble* images (left-most column) and corresponding clipped proposal abstracts (right-most column). The observation cycle and corresponding year, as well as proposal ID, are shown in the second and third columns, respectively. The proposal IDs link to the Mikulski Archive for Space Telescopes (MAST) page corresponding to the proposal.

### 2.1  *Hubble* Data Selection and Pre-processing

Observations corresponding to individual proposal IDs are queried through the Mikulski Archive for Space Telescopes (MAST)[2] via the *Astroquery* (Ginsburg et al., 2019) API. Products of type PREVIEW are filtered in, corresponding to preview postcard images. We note that these are not science-grade observations, but rather lower-resolution images useful for diagnostic or preview purposes. A maximum of 20 images are downloaded per proposal ID, selected at random, in order to avoid biasing the model towards proposals with a larger number of observations and survey-style campaigns. Images are centered and resized to a resolution-per-side of 512 pixels. Color previews (i.e., observations taken with

---

[1] https://archive.stsci.edu/hst/proposal_abstracts.html
[2] https://mast.stsci.edu/

| Prop. ID | LLM-extracted summary | |
| --- | --- | --- |
| | **Objects and phenomena** | **Science use cases** |
| 7340 | young oxygen-rich supernova remnants SN0540–69.3, LMC, SMC, supernova debris, active pulsar, synchrotron nebula | characterize ionization structure and distribution of chemically peculiar debris in SN0540–69.3, determine ionization structure in the SN debris of E0102.2–7219, provide benchmarks for models of nucleosynthesis in massive stars, excitation mechanisms in extremely metal-rich plasmas, and supernova explosion dynamics, study the pulsar and synchrotron nebula in SN0540–69.3, investigate SN0540–69.3's proximity to SN 1987A in both space and time, and relation to the same extended complex of young stars |
| 13757 | type Iax supernovae, white dwarfs, possible companion stars, accretion disks, luminous blue stars | constrain progenitor systems of type Iax supernovae, distinguish between explosion mechanisms, investigate mass transfer processes in accretion disks, determine if type Iax supernovae originate from massive stars |

Table 2: For the *Hubble* proposal abstracts shown in Tab. 1, the LLM (MIXTRAL-8X7B)-extracted summaries showing objects and phenomena (middle column) as well as potential downstream science use cases (last column) separately. The proposal IDs (left column) contain hyperlinks to the MAST page corresponding to the proposal.

multiple wavelength filters assigned to individual RGB channels) are manually excluded via a filename filter in order to maintain consistency across the samples. If no appropriate images corresponding to an abstract are found, it is excluded from the dataset.

In total, 31,859 images corresponding to 4,438 abstracts are included in the fine-tuning dataset. 3,194 images are held out for validation, with no abstract being common between training and validation sets in order to ensure an independent set of image-text pairs for evaluation. The held out images correspond to 429 unique abstracts. Due to practical limitations associated with the small size of the fine-tuning dataset, we did not use different datasets for validation and testing, deeming the current approach sufficient for a proof-of-principle exposition.

We note that some fraction of the image-caption pairs in the constructed dataset will primarily concern instrumentation and/or calibration rather than scientific content. We choose to not filter out these pairs, in order to have a larger sample of HST observations that the model can leverage to adapt to the distinctive characteristics of *Hubble* images.

## 2.2 Abstract Summarization via Guided Generation

Raw proposal abstracts summarize the corresponding successful HST observing proposals, which intend to make the case for allocating *Hubble* telescope time towards a particular set of observations. These abstracts are written in a diversity of styles, formats, and lengths while also being highly variable in their content. Although the abstracts can be used as-is as image captions, we experiment with summarizing them via guided large language model (LLM) generation to standardize the captions used for fine-tuning the CLIP model. Captions are summarized by extracting a list of objects and phenomena, as well as potential downstream science use cases, corresponding to the eventual imaged observation. The intended goal of the summarization process is to increase the strength of the association signal between text and images.

The method from Willard & Louf (2023) is used to produce an LLM-generated summary of the abstract conforming to a particular schema, specified in JSON format. The schema is designed to represent a list of the objects (e.g., 'Type Ia supernova') and phenomena

(e.g., 'gravitational lensing'), as well as potential downstream science uses cases (e.g., 'set constraints on supernova explosion models') that could correspond to the eventual imaged observation given the abstract text, with a minimum of 1 and a maximum of 5 elements per list. The procedure guides the generation of LLM outputs while ensuring that the schema is respected at every step in the generation process by masking out tokens that would violate the intended format. By framing the problem in terms of transitions between a set of finite states (i.e., a finite-state machine), Willard & Louf (2023) showed that guided generation can be performed with negligible overhead compared to unconstrained generation. See App. A.1 for a more detailed description of the guidance generation method used here, including an overview of technical details. While the schema-guided generation ensures the *format* of the output, the prompt and choice of LLM will dictate the *content* of the generated summaries. We use the open-weights, instruction-tuned model MIXTRAL-8X7B-INSTRUCT (Jiang et al., 2024) to generate the summaries, with guided generation performed using the *Outlines*[3] package. Further details on the summarization procedure, including the prompts and schema used, are provided in App. A.2.

Examples of LLM-generated abstract summaries are shown in Tab. 2, for the same set of abstracts as shown in Tab. 1. We train separate models using the raw abstracts and the LLM-generated summaries, and compare their performance on downstream tasks in Sec. 4. We note that, even after summarization, the association signal is expected to be noisy, since parts of the summarized caption may not be directly descriptive of the observed images. The goal of the fine-tuning process is to leverage the signal contained in this noisy association.

## 3 Methodology

### 3.1 Contrastive Language-Image Pre-training

Contrastive Language-Image Pre-training (CLIP; Radford et al., 2021) is a multi-modal neural network model pre-trained on a large corpus of image-text pairs via weak supervision using a contrastive loss. Given a minibatch $\mathcal{B}$ of $|\mathcal{B}|$ image-text pairs $\{(I_i, T_i)\}$, the goal is to align the learned representations of corresponding (positive) pairs $(I_i, T_i)$ while repelling the representations of unaligned (negative) pairs $(I_i, T_{j \neq i})$. Image and text encoders $f : I \to \mathbb{R}^{n_{\text{emb}}}$ and $g : T \to \mathbb{R}^{n_{\text{emb}}}$ are used to map images and text to a common embedding space of dimension $n_{\text{emb}}$. We use the standard bidirectional variant of the InfoNCE (Oord et al., 2018) contrastive loss function introduced for training CLIP-style architectures (Radford et al., 2021),

$$\mathcal{L}(\mathcal{B}) = -\frac{1}{2|\mathcal{B}|} \sum_{i=1}^{|\mathcal{B}|} \left( \log \frac{e^{x_i \cdot y_i / \tau}}{\sum_{j=1}^{|\mathcal{B}|} e^{x_i \cdot y_j / \tau}} + \log \frac{e^{x_i \cdot y_i / \tau}}{\sum_{j=1}^{|\mathcal{B}|} e^{x_j \cdot y_i / \tau}} \right), \text{ where } x_i = f(I_i) / \|f(I_i)\|$$

and $y_i = g(T_i) / \|g(T_i)\|$ are the normalized representations of the $i$-th image and text caption, respectively, and $\tau$ is a learnable temperature hyperparameter. Note that this loss treats the image and text representations symmetrically, ensuring that the two modalities are considered on the same footing.

### 3.2 Fine-tuning Procedure

The base CLIP model is fine-tuned using the dataset described in Sec. 2, using either the LLM-summarized abstracts or raw proposal abstracts paired with observations. When using raw proposal abstracts, random chunks of the text delimited by periods are selected on the fly to fit within the maximum token length of the text encoder. Images are augmented via random four-fold rotations (increments of 90°) and randomly cropped to the native resolution of the image encoder, maintaining $\sim 20\%$ of the area of the original image, at each training step. Given the relatively modest size of the fine-tuning dataset, a batch size $|\mathcal{B}| = 32$ is used throughout; larger batch sizes were observed to be susceptible to overfitting. The temperature hyperparameter $\tau$ was initialized to its pre-trained value. We emphasize that the positive and negative image-text association is noisy and imperfect, since multiple images can be associated with the same abstract, and the goal of the fine-tuning process is to leverage the signal contained in this noisy association.

---

[3]https://github.com/outlines-dev/outlines

We use the `CLIP-ViT-B/16` (Radford et al., 2021) variant as the base pre-trained CLIP model. We explore three different methods of training the model on our domain dataset: *(1)* Fine-tuning the entire network starting from the pre-trained base model; *(2)* Freezing the base image/text encoders and training a small projection head; and *(3)* Training the entire model from scratch. For *(2)*, we use a 2-layer MLP with 1024 hidden units and a GELU activation layer, projecting onto the 512-dimensional common embedding space. Additional details on the CLIP model and fine-tuning procedure are provided in App. B.

### 3.3 Evaluation Metrics

The model is evaluated by tracking the contrastive loss as well as the top-$k$% retrieval accuracy on the held out validation set over the course of training. The retrieval accuracy is defined as the fraction of associated captions (either raw or LLM-summarized abstracts) which fall within the top $k$% of captions by cosine similarity of the normalized image and caption embeddings, averaged over the images in the validation set: $\frac{1}{|\mathcal{V}|} \sum_{i=1}^{|\mathcal{V}|} \mathbb{1} \left[ \text{rank} \left( x_i \cdot y_i; \{x_i \cdot y_j\}_{j=1}^{|\mathcal{V}|} \right) \leq \left\lfloor \frac{k}{100} |\mathcal{V}| \right\rfloor \right]$ where $|\mathcal{V}|$ is the total number of images in the validation set, $\mathbb{1}[\cdot]$ is the indicator function that returns 1 if the condition inside the brackets is true and 0 otherwise, $\text{rank} \left( x_i \cdot y_i; \{x_i \cdot y_j\}_{j=1}^{|\mathcal{V}|} \right)$ is a function that returns the rank of the cosine similarity between $x_i$ and $y_i$ among the cosine similarities between $x_i$ and all captions $y_j$ in the validation set, and $k$ is the percentage of top captions considered for the retrieval accuracy. Note that this metric is symmetric in the image and text modalities.

We also qualitatively evaluate the learned embeddings through image retrieval (i.e., retrieving the most relevant images from the validation set using natural language queries).

## 4 Results and Discussion

### 4.1 Quantitative Evaluation

**Validation metrics during training** Figure 2 shows the contrastive loss (left) and the top-10% retrieval accuracy (right) evaluated on the held out validation set over the course of training, for different training configurations considered. The dashed orange lines show the metrics evaluated when training with batches where the image-text associations are randomly shuffled. This randomized baseline is seen to do on par with random expectation (i.e., a 10% retrieval accuracy), unlike the others, validating the presence of a significant association signal between images and text in the dataset. Interestingly, the base pre-trained model performs better than random expectation, with a top-10% retrieval accuracy of $\sim 15$% (as see from the left-most datum in Fig. 2 right, for the curves corresponding to fine-tuned models). We therefore also compare the qualitative performance of the base model with the fine-tuned models on downstream retrieval tasks.

The model trained using LLM-summarized abstracts (red lines) is seen to perform slightly worse than the model using raw abstracts as captions (blue lines), despite the curation of the summarized-abstract dataset intended to provide a stronger image-text association signal. Fine-tuning a small MLP head over frozen vision and text backbones (dotted green lines) and training from scratch with summarized abstracts as captions (yellow lines) show a non-trivial improvement compared to the base model, although with deteriorated performance compared to fine-tuning with either summarized or raw abstracts.

**Distribution of text-image cosine similarities** Figure 3 (left) shows the distribution of cosine similarities between corresponding image and text embeddings, $x_i$ and $y_i$, for the base CLIP model (purple line), and for the LLM-summarized abstracts using the fine-tuned CLIP model (red line). Distributions evaluated for a shuffled order of text embeddings – therefore randomizing the image-text correspondence during evaluation – are shown as dashed lines. We note that the shuffling here is performed at the evaluation stage, and not the training stage. The distributions for the base model is seen to be sharply peaked at a specific value, showing little diversity and being very similar between the shuffled (dashed

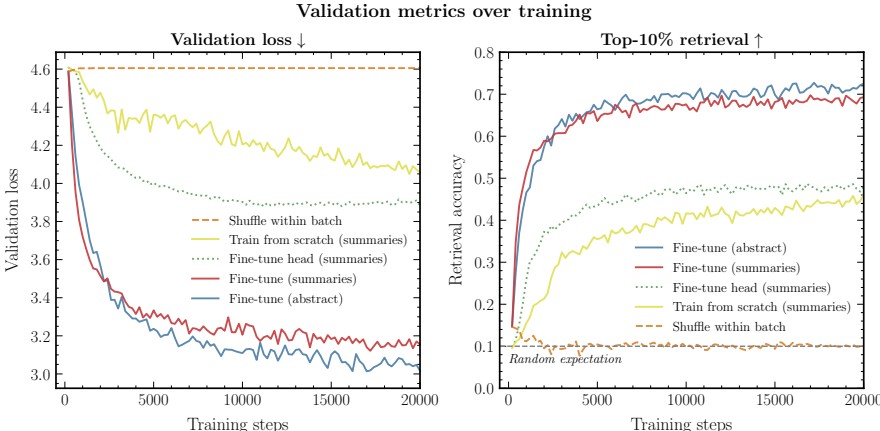

Figure 2: The CLIP contrastive loss (left) and the top-10% retrieval accuracy (right) computed on the validation set over the course of training. Shown for the dataset with summarized abstracts as captions (red), dataset using raw proposal abstracts as captions (blue), only fine-tuning a small MLP head (dotted green), training from scratch with summarized abstracts as captions (yellow), and trained with shuffled image-text pairs (dashed orange).

purple) and non-shuffled (solid purple) versions. Distributions for the fine-tuned model, on the other hand, show a clear separation when evaluated on shuffled (dashed red) and corresponding (solid red) text-image pairs.

**Retrieval accuracy** Figure 3 (right) shows the retrieval accuracy as a function of the retrieval fraction $k$%. In this case, we evaluate all four models (fine-tuned on raw abstracts (blue), fine-tuned on LLM-summarized abstracts (red), trained on LLM-summarized abstracts from scratch (yellow), and the base model (purple)) on the same captions dataset – the summarized abstracts – for a direct comparison. Remarkably, the model trained on raw abstracts shows very similar performance when evaluated on the summarized abstracts compared to that trained on the summarized abstracts themselves, indicating that *(1)* the image-text association signal is preserved in the summarization process, and *(2)* the model is able to effectively leverage meaningful concepts in the noisy raw abstracts through weak supervision. The significantly worse performance of the model trained from scratch, compared to the fine-tuned models, highlights the crucial role of the inductive bias inherited from the base pre-trained model, which effectively captures rich associations between images and language.

We show retrieval accuracy performance for additional variations on the model and training configuration in App. C.

## 4.2 Image Retrieval

Having aligned the image and text representations, we can embed a natural language query using the model and show the closest images by embedding from the validation set when ranked by cosine similarity. A sketch of this procedure is shown in Fig. 1 (right). We show these in Tabs. 3 and 4 for the base and fine-tuned models respectively using two simple curated queries: Jupiter and SN1987A (a specific supernova). The proposal ID corresponding to the retrieved images is shown below each image, and contains a hyperlink to the MAST page corresponding to the proposal for further details.

While the base model shows some signs of meaningful retrieval (e.g., the image of Jupiter in the first row of Tab. 3), associations between the retrieved images and corresponding queries are not consistent.

The model fine-tuned with summarized abstracts, meanwhile, shows strikingly different behavior (Tab. 4). Images looking like Jupiter are returned for the Jupiter query. However,

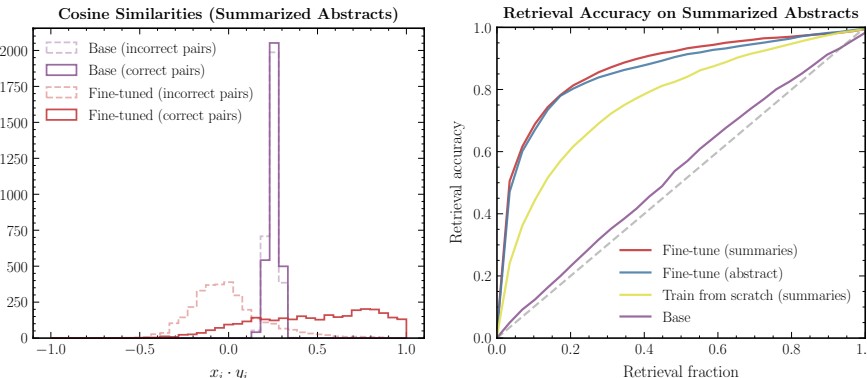

Figure 3: (Left) Distribution of cosine similarities between corresponding image and text embeddings, $x_i$ and $y_i$, shown when using the base CLIP model (purple lines), and the summary fine-tuned CLIP model (red line). Dashed lines correspond to models evaluated on image-text pairs with associations shuffled. (Right) Retrieval accuracy as a function of the retrieval fraction $k$ for the fine-tuned model on the summarized abstracts (red), fine-tuned on raw abstracts (blue), trained on summarized abstracts from scratch (yellow), and the base model (purple).

this example also illustrates the model's potential to misidentify objects, with the first and third image actually showing Saturn with artifacts on the planet and partially obscured rings. Supernova SN1987 itself can be seen in the three closest images for the SN1987A query.

We also evaluate the observation retrieval task more quantitatively. We design a prompt which lets us evaluate whether the abstract corresponding to a retrieved observation is relevant or not, with the output constrained to be a boolean using *Outlines*. We then evaluated this prompt, for the base as well as fine-tuned models, on the top 10 closest images by cosine similarity returned for 10 different queries. 38% and 77% of the retrieved observations are deemed relevant when using the base and fine-tuned models, respectively. The fine-tuned model is thus significantly more likely to return images relevant to the query. The prompt and curated queries for this test are described in App. A.3.

Note that we chose to illustrate qualitative performance on image retrieval using the model fine-tuned on summarized abstracts, rather than raw abstracts. Although the two models show very similar quantitative performance on retrieval metrics (as shown in Fig. 3), they exhibit characteristically different behaviors in terms of images retrieved, with the summary fine-tuned models generally retrieving images that look more visually "relevant" to a domain expert. We emphasize that for scientific usefulness, the goal is not necessarily to correctly retrieve the most relevant objects, but rather to identify a diverse set of interesting candidates for manual follow-up and further analysis. By diverse, we mean that retrieved observations may contain different types of objects or phenomena, which may be relevant to the query in distinct ways.

The fine-tuned model can similarly be used for description/text retrieval, akin to the traditional zero-shot classification setting, where the closest text snippets from a curated list are returned given an observed astronomical image. We show examples of the text retrieval task in App. D.

## 5 Outlook and Conclusions

We present PAPERCLIP, a method for training domain-specific multi-modal models for astrophysics that associates observations imaged by telescopes with natural language in a common embedding space. We showcase an application to *Hubble* Space Telescope (HST) observations, where the model is fine-tuned from a pre-trained CLIP model using abstracts of successful *Hubble* proposals, optionally summarized, leveraging a noisy association

| Query | Top-3 most similar images using base off-the-shelf CLIP model |
|---|---|

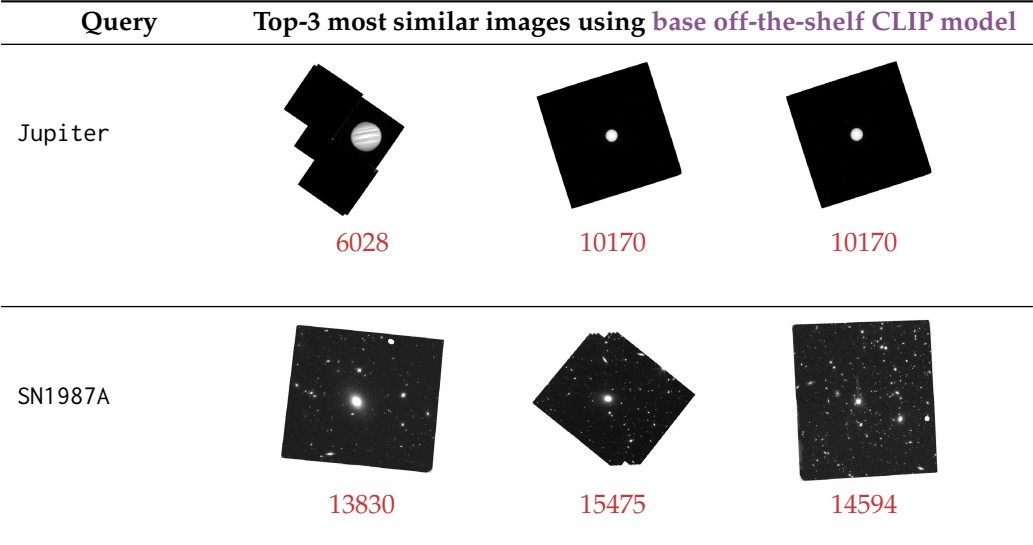

Table 3: For two text queries (left-most column), the three most similar images from the validation dataset by cosine similarity when using the **base (off-the-shelf) CLIP model** (`CLIP-ViT-B/16`). The proposal ID associated with each image is given below the image and contains a hyperlink to the MAST page corresponding to the proposal.

| Query | Top-3 most similar images using summary fine-tuned CLIP model |
|---|---|

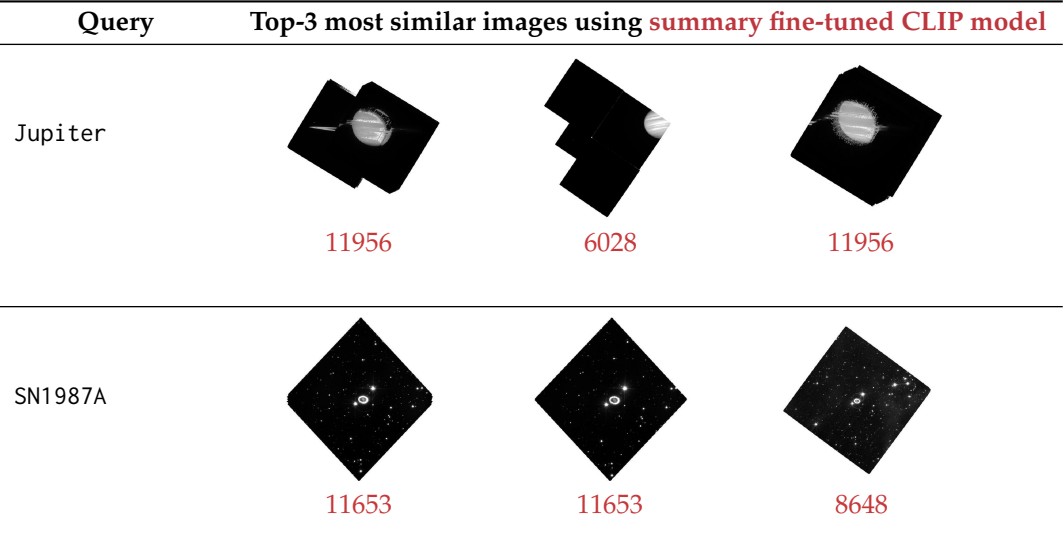

Table 4: Same as Tab. 3, but using the **summary fine-tuned CLIP model**.

signal between text and images. We show that PAPERCLIP significantly outperforms the base CLIP model in quantitative metrics, such as retrieval accuracy, as well as quality of text-to-image and image-to-text retrieval. We also introduce a novel LLM summarization process which leverages guided generation to distill the content of proposal abstracts while preserving salient information. Overall, the procedure demonstrates the efficacy of fine-tuning generalist pre-trained models on small amounts of domain-specific data, in particular astronomical datasets, and leveraging text as an interface for interacting with the data.

Although the model explored here is fine-tuned using postage stamp images (i.e., preview-quality and not science-grade data), we highlight potential immediate as well as downstream

use cases. A model trained using weakly-supervised image-text pairs can be used to query large amounts of unlabeled survey data e.g., PHANGS (Lee et al., 2022), COSMOS (Scoville et al., 2007) for objects or use-cases of interest using natural language, as well as to efficiently find patterns in such data that may not be apparent using specialized models or manual inspection. The learned representations, having shown to correlate with physical characteristics of imaged objects, can also be fine-tuned via transfer learning to adapt to either specific tasks e.g., classification (Wei et al., 2020) or segmentation (Hausen & Robertson, 2020), or observations imaged by other telescopes.

Finally, while the CLIP model is restricted to retrieving nearest-neighbour associations within and across text/image modalities, the learned embeddings can be used as a starting point for training or fine-tuning multi-modal large-language models for interacting with survey data and receiving responses in natural language form, as well as grounding the responses based on an existing set of observations.

### Acknowledgments

We thank Michael Brenner, François Lanusse, and Julian Muñoz for helpful conversations. This work is supported by the National Science Foundation under Cooperative Agreement PHY-2019786 (The NSF AI Institute for Artificial Intelligence and Fundamental Interactions, http://iaifi.org/). This material is based upon work supported by the U.S. Department of Energy, Office of Science, Office of High Energy Physics of U.S. Department of Energy under grant Contract Number DE-SC0012567. YS was supported by the Research Science Institute (RSI) program at MIT. This research was supported by an award from Google, "Interpretation of Multimodal Images from Astronomy". This research was supported by the Munich Institute for Astro-, Particle and BioPhysics (MIAPbP), which is funded by the Deutsche Forschungsgemeinschaft (DFG, German Research Foundation) under Germany's Excellence Strategy – EXC-2094 – 390783311. The computations in this paper were run on the FASRC Cannon cluster supported by the FAS Division of Science Research Computing Group at Harvard University.

This research is based on observations made with the NASA/ESA Hubble Space Telescope obtained from the Space Telescope Science Institute, which is operated by the Association of Universities for Research in Astronomy, Inc., under NASA contract NAS 5-26555. Based on observations made with the NASA/ESA Hubble Space Telescope, and obtained from the Hubble Legacy Archive, which is a collaboration between the Space Telescope Science Institute (STScI/NASA), the Space Telescope European Coordinating Facility (ST-ECF/ESAC/ESA) and the Canadian Astronomy Data Centre (CADC/NRC/CSA).

This work relied on the *Astroquery* (Ginsburg et al., 2019), *BitsAndBytes* (Dettmers et al., 2022), *Flax* (Heek et al., 2023), *Jax* (Bradbury et al., 2018), *Jupyter* (Kluyver et al., 2016), *Matplotlib* (Hunter, 2007), *Numpy* (Harris et al., 2020), *Optax* (Babuschkin et al., 2020), *Outlines*, *Pandas* (Virtanen et al., 2020), *Pydantic*, *PyTorch* (Paszke et al., 2019), *SciPy* (Virtanen et al., 2020), *Transformers* (Wolf et al., 2019), and *Wandb* (Biewald, 2020) software packages.

### Reproducibility Statement

Code used to reproduce the results in this work is available at https://github.com/smsharma/PAPERCLIP-Hubble/tree/main.

### Ethics Statement

This work relies on using abstracts from successful *Hubble* Space Telescope observing proposals as part of a dataset for training and evaluating machine learning models. While these abstracts are publicly available, the authors likely did not anticipate their text being used in this manner, raising questions around consent, attribution, and appropriate use of data. Since this research intends to develop methods to aid astronomical research and does not use sensitive personal information or target commercial gain, we believe that the scientific benefits outweigh the potential concerns in this case, while acknowledging good-faith arguments to the contrary. As the use of foundation models in the sciences increases,

it will be important for the community to consider norms and guidelines around the appropriate use and attribution of various data sources for model training and evaluation, including qualitative textual data, to ensure transparency and maintain trust.

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

# A    Details on the Abstract Summarization Procedure

We provide additional details of the abstract summarization procedure, including a brief overview of the guided generation method used, as well as the prompts and schema used for the summarization task.

## A.1    Guided LLM Generation with *Outlines*

As mention in Sec. 2.2, we employ the guided generation method introduced by Willard & Louf (2023) and implemented in *Outlines* to ensure that the LLM summarization of the raw proposal abstracts adheres to specific pattern, specified in JSON format (Sec. A.2 below), which we briefly describe here. This approach represents the desired output format as a finite-state machine (FSM) that encodes the JSON schema as a regular expression. The JSON schema constraint is therefore first converted into a regular expression.

The key idea then is to pre-compute an index that maps each state of the FSM to the subset of tokens from the LLM's vocabulary that can be generated from that state while still allowing for a valid completion of the pattern. By doing so, we can efficiently determine the valid next tokens at each step of the generation process without having to check the entire vocabulary.

Formally, let $\mathcal{M} = (Q, \Sigma, \delta, q_0, F)$ be the FSM representing the regular expression, where $Q$ is the set of states, $\Sigma$ is the alphabet of the regular expression, $\delta : Q \times \Sigma \to Q$ is the transition function between states, $q_0$ is the start state, and $F \subseteq Q$ is the set of accept states which terminate the generation. An index $\sigma : Q \to \mathcal{P}(V)$ is first constructed, where $V$ is the LLM's token vocabulary and $\mathcal{P}(V)$ denotes the power set of $V$. For each state $q \in Q$, $\sigma(q)$ contains the allowed tokens that can be generated from state $q$ while maintaining the possibility of reaching an accept state. The construction of $\sigma$ involves finding all token sequences that, when processed by the FSM starting from each state $q$, lead to an accept state.

During the sequential generation process, the current FSM state $q_t$ is kept track of after sampling each token $v_t$. At each step $t$, the LLM's output logits are masked based on the valid next tokens $\sigma(q_t)$, setting the logits of invalid tokens to $-\infty$. The next token is then sampled from the categorical distribution defined by the unmasked logits, and the FSM transitions to the next state $q_{t+1} = \delta(q_t, v_{t+1})$, where $v_{t+1} \in \Sigma$ is the token in the regular

expression alphabet corresponding to the sampled token. This process continues until an accept state with no outgoing transitions is reached, indicating a valid completion of the pattern.

## A.2 Prompts and Schema Used for Summarization

We list here the prompts and schema (i.e., desired output formats) used for guided text generation via *Outlines* package interfacing with the MIXTRAL-8X7B-INSTRUCT open-weights large language model.

The following schema, specified using the data-validation package *Pydantic*, is used to guide the generation of the summaries, intended to produce between one and five objects and hypotheses, as well as science use cases, given a raw proposal abstract. Both fields are of type conlist, a *Pydantic* type that represents a constrained list.

```python
from pydantic import BaseModel, conlist

class ConstrainedResponseHST(BaseModel):
    objects_and_phenomena: conlist(str, min_length=1, max_length=5)
    science_use_cases: conlist(str, min_length=1, max_length=5)
```

The following prompt function is used to produce a list of one to five possible objects and phenomena shown in HST observations downstream of a proposal abstract, as well as one to five possible science use cases, in the format native to *Outlines*. `"[INST]"` and `"[/INST]"` are start and end instruction delimiters, respectively, for the MIXTRAL-8X7B model.

```python
import outlines

@outlines.prompt
def prompt_fn(abstract):
    """"[INST] You are an expert astrophysicist, with broad expertise across
    observational and theoretical astrophysics. You are able to extract core
    information from astrophysical texts.

Abstract: "{{abstract}}"

Based on the above observational proposal abstract, your task is to summarize the
    nature of the eventual observations. You will identify the astrophysical
    objects and phenomena, as well as the potential science use cases described
    in the abstract.

Follow these instructions exactly:
- Mention up to 5 items for both categories; do not mention more than 5 items in
    either category.
- Choose the most relevant ones if there are more than 5 items in a category.
- Never mention the Hubble Space Telescope, HST, or the HST archive.
- Mention the class (e.g., barred spiral galaxy) and not just the specific
    instance (e.g., Andromeda).
- Name the objects in the science use cases, if appropriate.
- Write out full names of objects in addition to acronyms.
- Do not list irrelevant objects which do not describe the eventual observation,
    such as units or proposal Cycle numbers. List fewer but more relevant objects
    , if in doubt.
- Each science case listed must be self-contained but succinct.
- Only write in English.
- Do not list items that are too generic (e.g., galaxy, faint object, kinematics)
- The total length of text should not exceed 80 words.
- Present your lists in a comma-separated format; no dashed or numbered lists.

Example output: {'objects_and_phenomena':'spiral galaxies, galaxy clusters,
    supernova remnants', 'science_use_cases':'model galactic structure and
```

```
         evolution, characterize dark matter distribution in clusters, analyze
         expansion rates of supernova remnants'}
26
27  Answer in JSON format. The JSON should be a dictionary with keys "
         objects_and_phenomena" and "science_use_cases".
28
29  [/INST]
30  """
```

## A.3 Prompt Used for Quantitative Evaluation of Observation Retrieval

The following prompt was used to evaluate the relevance of abstracts corresponding to retrieved images to a query, when quantitatively assessing the observation retrieval task.

```
1   import outlines
2
3   @outlines.prompt
4   def prompt_fn(abstract, query):
5       """[INST]
6   You are an expert astrophysicist, with broad expertise across observational and
         theoretical astrophysics.
7
8   Abstract: "{{abstract}}"
9   Query: "{{query}}"
10
11  The above is an abstract for a proposed observation taken by the Hubble Space
         Telescope (labeled "Abstract"), and an object or concept (labeled "Query").
12
13  Could the observations corresponding to the abstract contain the query? Be
         precise, and do not contain related concepts or objects.
14
15  Your response should be either True or False. Only return True if the query is
         closely related to the abstract, and the downstream observation could be
         relevant to the query.
16  [/INST]
17  """
```

The queries used in the evaluation were ["globular cluster", "dwarf galaxy", "SN1987A", "strong lensing", "galaxy clusters", "interstellar medium", "dark matter", "spiral galaxies", "lyman alpha", "comets"].

# B   Additional Model and Training Details

We use the CLIP-ViT-B/16 (Radford et al., 2021) variant as the base pre-trained CLIP model. This model uses a 12-layer, 12-head, 768-embedding dimension vision transformer with patch size $16 \times 16$ as the image encoder (Dosovitskiy et al., 2020) and a 12-layer, 8-head, 512-embedding dimension text sequence transformer as the text backbone (Vaswani et al., 2017). The text encoder has a maximum length of 77 tokens and the image encoder has a native resolution of $224 \times 224$ pixels. Linear projection layers map the outputs of the image and text encoders to a common embedding space of dimension $n_{emb} = 512$. In total, the model contains $\sim 149$ million trainable parameters. This model was originally pre-trained on $\sim 400$ million image-text pairs from internet data.

All models were trained over 20,000 steps with 2000 linear warmup steps using the AdamW optimizer (Loshchilov & Hutter, 2019; Kingma & Ba, 2015) with learning rate $10^{-5}$ and weight decay $10^{-3}$. Training takes approximately 3 hours on 4 Nvidia A100 GPUs. Models were instantiated using the *Transformers* (Wolf et al., 2019) library and trained using packages from the *Jax* (Bradbury et al., 2018) ecosystem.

## C  Additional Variations on Model and Training

Figure 4 shows the retrieval accuracy as a function of the retrieval fraction for further variations of the model or training, evaluated and trained on summarized abstracts. The red line corresponds to the model trained on summarized abstract described in the main text (fine-tuned on `CLIP-ViT-B/16` with constant learning rate $LR = 10^{-5}$ after linear warmup). The purple line corresponds to the base `CLIP-ViT-B/16` model.

Curves for the model fine-tuned on the larger base CLIP model `CLIP-ViT-L/14` (dotted red), with a smaller learning rate $LR = 10^{-6}$ (dashed green), and with a cosine learning rate schedule (green) are also shown. All these models are seen to perform similarly, with the exception of the model trained with smaller learning rate showing degraded performance. Given the similar performance between `CLIP-ViT-L/14` ($\sim 428$ million parameters) and `CLIP-ViT-B/16` ($\sim 149$ million parameters), we chose the latter as the base model in the main text for computational efficiency.

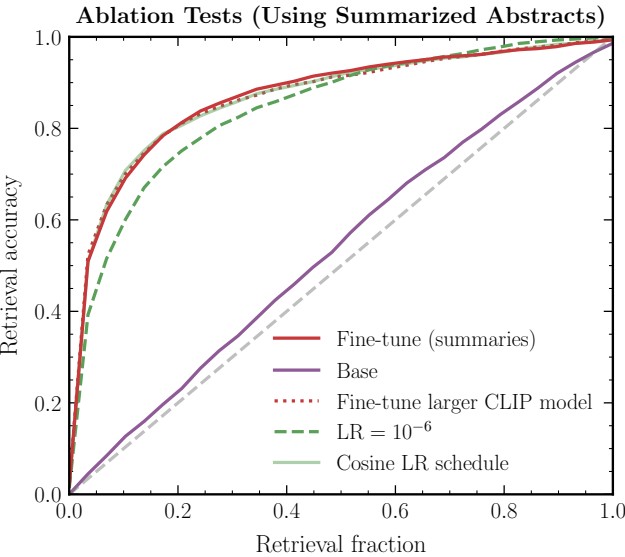

Figure 4: Same as Fig. 3 (right) – retrieval accuracy as a function of the retrieval fraction – for further variations on the model or training. The red and purple lines correspond to the model trained on summarized abstract, described in the main text, and the base `CLIP-ViT-B/16` model, respectively. Curves for the model fine-tuned on the larger base CLIP model `CLIP-ViT-L/14` (dotted red), with a smaller learning rate $LR = 10^{-6}$ (dashed green), and with a cosine learning rate schedule (green) are also shown.

## D  Text Retrieval Task

We can use images from the validation set as queries and retrieve the most relevant text chunks (e.g., objects and use cases) from a curated list.

The following curated categories are used in the text retrieval experiment in Sec. 4. These are derived by initially prompting CLAUDE 2[4], having attached a subsample of 30 proposal abstracts in the online interface to be used as context, to produce a list of categories corresponding to typical HST observations. The list is then manually curated to remove similar entries and ensure a representative sample of categories.

---

[4]https://claude.ai/

```
1  ["star forming galaxies", "lyman alpha", "dust", "crowded stellar field", "core-
       collapse supernova", "cosmology", "gravitational lensing", "supernovae", "
       diffuse galaxies", "globular clusters", "stellar populations", "interstellar
       medium", "black holes", "dark matter", "galaxy clusters", "galaxy evolution",
        "galaxy formation", "quasars", "circumstellar disks", "exoplanets", "Kuiper
       Belt objects", "solar system objects", "cosmic web structure", "distant
       galaxies", "galaxy mergers", "galaxy interactions", "star formation", "
       stellar winds", "brown dwarfs", "white dwarfs", "nebulae", "star clusters", "
       galaxy archeology", "galactic structure", "active galactic nuclei", "gamma-
       ray bursts", "stellar nurseries", "intergalactic medium", "dark energy", "
       dwarf galaxies", "barred spiral galaxies", "irregular galaxies", "starburst
       galaxies", "low surface brightness galaxies", "ultra diffuse galaxies", "
       circumgalactic medium", "intracluster medium", "cosmic dust", "interstellar
       chemistry", "star formation histories", "initial mass function", "stellar
       proper motions", "binary star systems", "open clusters", "pre-main sequence
       stars", "protostars", "protoplanetary disks", "jets and outflows", "
       interstellar shocks", "planetary nebulae", "supernova remnants", "red giants"
       , "Cepheid variables", "RR Lyrae variables", "stellar abundances", "stellar
       dynamics", "compact stellar remnants", "Einstein rings", "trans-Neptunian
       objects", "cosmic microwave background", "reionization epoch", "first stars",
        "first galaxies", "high-redshift quasars", "primordial black holes", "
       resolved binaries", "binary stars"]
```

The following prompt is used to generate the initial list before manual curation: *"Here is a list of Hubble proposals. Base on this, please provide a list of about 100 strings, each describing a science target or use case for observations imaged by the Hubble Space Telescope. You may use these proposals and also rely on your general knowledge. For example, ["gravitational lensing", "supernovae", "diffuse galaxies", ...]"*

We show the result of image-to-text retrieval in Tab. 5, for the base (second column) as well as summary fine-tuned (third column) models, using four observations (left-most column) from the validation set.

The top four text associations are shown for each image query. The 'ground truth' summarized abstract is shown in the right column. The base as well as fine-tuned models are seen to return a mix of relevant and less-relevant associations, although showing different qualitative behavior. Purely qualitatively, the fine-tuned model is seen to consistently return more relevant associations compared to the base model.

The second row (an image of supernova 1987A) highlights an interesting pattern – the base model erroneously attributes the object at the center of the image to a gravitational lens, while the fine-tuned model correctly identifies it as a supernova remnant. This kind of reasonable misattribution is common when querying the base model, and largely absent in the fine-tuned model.

| *Hubble* image | Top-4 text (base off-the-shelf) | Top43 text (summary fine-tuned) | Summarized abstract (objects; 'ground truth') |
|---|---|---|---|
|  | 1. high-redshift quasars
2. gravitational lensing
3. white dwarfs
4. dwarf galaxies | 1. dwarf galaxies
2. RR Lyrae variables
3. red giants
4. trans-Neptunian objects | isolated dwarf galaxies, WLM, Pegasus Dwarf Irregular Galaxy, stellar mass, main sequence stars |
|  | 1. gravitational lensing
2. supernovae
3. binary star systems
4. circumstellar disks | 1. supernova remnants
2. protostars
3. galactic structure
4. core-collapse supernova | supernova SN 1987A, circumstellar ring, supernova remnant, shocked ring, radioactive isotopes |
|  | 1. gravitational lensing
2. high-redshift quasars
3. ultra diffuse galaxies
4. galaxy clusters | 1. galaxy clusters
2. lyman alpha
3. intracluster medium
4. dark energy | X-ray luminous galaxy clusters, eMACS clusters, Balmer Break Galaxies, Lyman-break galaxies, gravitational telescopes |
|  | 1. star clusters
2. globular clusters
3. open clusters
4. stellar populations | 1. globular clusters
2. star clusters
3. galactic structure
4. crowded stellar field | pre-main sequence stars, Large Magellanic Cloud, young clusters, color-magnitude diagrams, main-sequence turn offs |

Table 5: Text snippets from a curated list most closely matching a given image query (leftmost column) by cosine similarity of respective embeddings, shown for the base off-the-shelf (CLIP-ViT-B/16) and summary fine-tuned models. The 'ground truth' LLM-summarized abstract (only objects/phenomena) is shown in the right-most column.

