# OpenReview forum: "PAPERCLIP: Associating Astronomical Observations and Natural Language with Multi-Modal Models"
_colmweb.org/COLM/2024/Conference — COLM_

### Official Review · Reviewer_kBem · 2024-05-03

**Rating:** 7
**Confidence:** 4
**Ethics Flag:** 1

**Summary:**

The paper deals with finetuning of the CLIP model for the astronomical data, using free-form proposals for the Hubble telescope as an textual input and images obtained for each proposal as an image input. The results show that finetuning results in meaningful image and text embeddings that allow for the semantic image search using textual input.

**Questions To Authors:**

I do not understand why image-text association should be noisy, as it is mentioned in the first paragraph in 3.2. InfoNCE allows for multiple positive pairs, so it is possible to train model taking all image-text correspondences into account simultaneously.

**Reasons To Accept:**

The paper deals with interesting data, and the results are promising.

The paper uses meaningful evaluation and baselines.

The paper is well written and structured, easy to follow and understand.

**Reasons To Reject:**

The dataset used in this study is an exception in astronomical data, since images have meaningful textual counterparts. The dataset is also very small, comparing to vast massives of available astronomical data. So, the most crucial question is whether it is possible to use the model trained on the Hubble proposal dataset to label/search some other available images. It would be nice to see at least qualitative evaluation in that regard.

I understand that it is appealing to call a model "PAPER CLIP" but this name is misleading: First, this abbreviation is not very informatie, it does not even mention anything astronomical. Second, and most importantly, the paper does not present any new model, it only fine tune the existing one. I would call the paper "Fine-tuning CLIP for matching astronomical images and texts", or something like this, without introducing any additional abbreviations.

I am not sure training with random batches is actually needed. Just random baseline and non-tuned CLIP would be enough in my view.

---

> ### Author Rebuttal · Authors · 2024-05-29
>
> **I understand that it is appealing to call a model "PAPER CLIP" but this name is misleading**
>
> We note that we do not call the _model_ PAPERCLIP; instead, as emphasized in the abstract and elsewhere, PAPERCLIP refers to the _method_ of using (processed) proposal abstracts to inform observation-text associations for astronomical images, which in this specific case we apply to observations from the Hubble Space Telescope.
>
> **The dataset used in this study is an exception in astronomical data, since images have meaningful textual counterparts.**
> **The most crucial question is whether it is possible to use the model trained on the Hubble proposal dataset to label/search some other available images.**
>
> The dataset includes diverse images from Hubble, with different filters, targets, and post-processings, taken over a ~30 year timespan. Our method is intended to be applied to observations from a specific telescope, and we do not expect it to immediately generalize to others without further fine-tuning. We emphasize that not all Hubble images have associated text i.e., not all are the downstream result of targeted proposals, as those used in fine-tuning here (the data used is a subset of Hubble observations). Our method is therefore intended to be applied to unlabeled sets of observations. In the paper, in order to be able to compare with a ground truth, we tested our method on observations which do have associated abstracts.
>
> **[...] why image-text association should be noisy [...] InfoNCE allows for multiple positive pairs, so it is possible to train model taking all image-text correspondences into account simultaneously.**
>
> Although the version of InfoNCE used in the paper allows for a batch of $N$ positive pairs, each element of the pair has $N-1$ negatively paired elements; a single data point cannot have multiple positive “anchors”. Therefore, since multiple images can correspond to an abstract, when training with minibatches it is possible that some of the negative pairs will actually be positively associated (e.g., if two abstracts in a batch are the same, but the corresponding images are different). Through the results in the paper, we empirically show that, despite this, the model is able to leverage the signal between positively associated pairs and “beat down” the noise associated with negatively pairing image-text samples that may nevertheless be associated.

---

### Official Review · Reviewer_Uywh · 2024-05-07

**Rating:** 5
**Confidence:** 4
**Ethics Flag:** 1

**Summary:**

**Quality:** The authors compile a dataset of textual proposals for astronomical observations written by astronomers so that they can perform these observations with the Hubble Space Telescope,  and the ensuing images observed by telescope. Using this dataset, the authors explore for the first time the original and interesting use case of image and text retrieval in the astronomy domain. The authors fine-tune CLIP using proposal abstracts and related observation images following CLIP’s contrastive learning approach. The authors also experiment with optional summarization of abstracts via LLM-guided summarization into a required format, prompting Mixtral to generate the summaries. The experiments performed are adequate and seem sound with improved results going beyond the zero-shot CLIP model and other baselines the authors compare to.

**Clarity:** The paper is mostly clear; however, certain methodological choices and domain-specific terminology related to astronomy should be explained in more detail to clarify the task and the inputs.

**Originality:** The application of pretrained models in the astronomy domain for images and texts is interesting and novel.

**Significance:** The main contribution is the investigation of the use of pretrained models in the domain of astronomy, revealing multimodal use cases in sciences that can benefit from the advancements in pretrained models. I think the outcomes would be of significance to many other scientific domains.

**Questions To Authors:**

1. It seems that the evaluation and the analysis was done on the validation set, did the authors have a separate test set?

2. The images are cropped to the 20% of the area of the original image, would this somehow get rid of informative parts in an image?

3. Why is the random expectation of retrieval accuracy 10%? In 3.3, the formula has the k% of the number of images, but the explanation mentions top k% of captions, which I found confusing. It would also be better to clarify which retrieval task the graph on the right of Figure 2 corresponds to.

4. In Table 3, the query is ‘Jupiter’ and the first retrieved image is of Jupiter, so I was not sure what the authors meant by ‘it is challenging to discern meaningful, strong associations’. Additionally, the authors claim that the fine-tuned model ‘shows strikingly different behavior’, which, as a non-expert for the astronomy domain, was not clear to me. Is it also because the authors used shorter/simpler queries as compared to the fine-tuning data?

5. The paper could benefit from having an excerpt of the prompts and schema samples in the main part, which I found informative.

**Reasons To Accept:**

The application of pretrained models to the astronomical domain is original, showing the applicability and usefulness of pretrained models in the sciences, especially on data that does not correspond to the large bulk of the pretraining datasets, originating from a specific distribution.

**Reasons To Reject:**

The paper is written mostly well, but I have some doubts regarding the methodological choices (no test set, cropping images, interpretation of examples; see my questions below), which would need more motivation and clarification in the main paper. Additionally, some domain-specific terminology needs to be explained, especially earlier in the paper. For instance, what is meant by ‘a successful abstract proposal’? This was explained halfway through the paper, which I think should be introduced earlier.

---

> ### Author Rebuttal · Authors · 2024-05-29
>
> **[...] did the authors have a separate test set?**
>
> Due to practical limitations associated with the small size of the fine-tuning dataset, we did not use different datasets for validation and testing, deeming this sufficient for a proof-of-principle exposition.
>
> **The images are cropped to the 20% of the area of the original image, would this somehow get rid of informative parts in an image?**
>
> The images are cropped to 20% on the fly during training, as a form of data augmentation. Since each image is seen ~20 times over training, each time cropped differently, the model can learn to leverage a positive association signal when this is present. The association signal may also not be localized to a particular part of the image.
>
> **Why is the random expectation of retrieval accuracy 10%? In 3.3, the formula has the k% of the number of images, but the explanation mentions top k% of captions [...]**
>
> The random retrieval accuracy is 10% because, without a learned association between modalities, the model will correctly retrieve the correct item in the top-k%, k% of the time by random chance.
>
> As mentioned in Sec. 3.3, the retrieval evaluation metric when averaged over the test set is symmetric over the image and text modalities; i.e., due to commutativity of the cosine similarity, we can either average over images in the test set while looking at top-k% captions, or vice-versa. In Sec. 3.3, for concreteness we describe text retrieval (“top k% of captions”, “averaged over the images”). Fig. 2 right therefore does not correspond to a particular retrieval task.
>
> **[...] I was not sure what the authors meant by “it is challenging to discern meaningful, strong associations”**
>
> While occasional associations are apparent, this is very much the exception rather than the norm for the base model (as can be seen from subsequent retrieved images); however, we agree that the language here should be softened.
>
> **[...] the authors claim that the fine-tuned model “shows strikingly different behavior’’, which, as a non-expert for the astronomy domain, was not clear to me.**
>
> The difference in behavior can be seen from the retrieved images – in Tab. 4 all the returned images for the fine-tuned model correspond to Jupiter and SN1987A, as appropriate. This is not the case for the base model, in Tab. 3. While domain expertise may be helpful in assessing relevance, one can click through to the Proposal IDs of the retrieved images, confirming their relation to the queries.

---

> > ### Comment · Reviewer_Uywh · 2024-06-04
> >
> > Thank you for the response. The idea behind image cropping is clear to me now. If each image is seen multiple times during training in this way, would that affect the chance of retrieving 'a correct item'? Is that also the case in the validation set? I think the authors should explicitly address in the paper that they do not use different datasets for validation and testing. I know that cosine similarity is symmetric, but it is not clear to me how the ranks are also symmetric in this setup. I think Section 3.3 should contain more clarifications. Is |V| the same for the number of images and the number of captions?

---

### Official Review · Reviewer_1Yks · 2024-05-09

**Rating:** 3
**Confidence:** 4
**Ethics Flag:** 1

**Summary:**

The paper presents an astronomical dataset of image-text pairs for finetuning CLIP. The dataset consists of 32k training and 3k validation images (paired with text). The dataset is constructed by taking abstracts of (funded!) proposals for the Hubble Space Telescope. The paper tries summarizing these abstracts with LLMs. Both raw and summaries work comparably in loss and retrieval performance on the validation split. The images corresponding to an abstract (up to 20) are found by querying an API.

**Reasons To Accept:**

- A cool dataset constructed from public resources. The dataset and the finetuned model can be useful for applications in astronomy.

**Reasons To Reject:**

- There is no technical novelty in data construction or modeling. The dataset is constructed by downloading abstracts and matching with images. The model is standard CLIP. The experiments on the constructed validation set are as expected.
- Only a loose connection with LLMs. The paper is primarily about CLIP.

---

> ### Author Rebuttal · Authors · 2024-05-29
>
> **There is no technical novelty in data construction or modeling. The dataset is constructed by downloading abstracts and matching with images. The model is standard CLIP. The experiments on the constructed validation set are as expected.**
>
> **Only a loose connection with LLMs. The paper is primarily about CLIP.**
>
> We frame our study as an application of language models to a previously-unexplored field with domain-informed data curation. We note that the COLM call mentions, “LMs on diverse modalities and novel applications [...] with extra encouragement for less studied modalities or applications.”
>
> Regarding the paper primarily using CLIP and having a loose connection with LLMs, we again note that call for papers mentions, “We consider the term "language model" in the broadest way.”, and is thus not restricted to LLMs specifically.
>
> Overall, we believe that an LM application in astronomy, which is an understudied field in the context of LMs with nevertheless high potential, is in-scope for COLM.

---

### Official Review · Reviewer_5XaM · 2024-05-13

**Rating:** 6
**Confidence:** 4
**Ethics Flag:** 2

**Summary:**

This paper introduces PAPAERCLIP, a fine-tuned version of the CLIP model designed to link proposal abstracts with their corresponding astronomical observations. Various fine-tuning methods and text processing strategies are explored. The authors demonstrate that their adapted model performs effectively in both image and description retrieval tasks on the HSB dataset.

**Questions To Authors:**

Given that the CLIP model is trained on a diverse but also unreleased dataset, is there some way to see how alinged the astronomical data with the CLIP model's domain? It is unclear whether the observed performance improvements are due to fine-tuning (from a generalist vision model) or if the original CLIP model had prior exposure to similar astronomical data, enabling rapid adaptation.

The paper extensively discusses the use of LLM-generated abstracts, though this approach does not yield superior results. Can the authors delve deeper into why this might be?

**Reasons To Accept:**

The study investigates the application of image foundation models to astronomical observations, highlighting potential benefits for astrophysics.

Various training strategies are explored, including fine-tuning, using adapters, and training from scratch.

The authors also apply the method of Willard & Louf (2023), utilizing LLM-generated abstract summaries as training texts.

**Reasons To Reject:**

The paper does not include any baseline comparisons. Questions arise such as: 1), Are there existing tools capable of performing similar tasks? 2) Assuming each observation has an abstract, could description/image retrieval be achieved by calculating text similarities (tfidf or neural network based), treating abstracts as a form of image representation to potentially establish an upper bound?

The exploration of different CLIP variants like OpenCLIP, and other similar models like SigLIP, could provide insight into the impact of varying backbone foundation models.

---

> ### Author Rebuttal · Authors · 2024-05-29
>
> **The paper does not include any baseline comparisons. [...] 1), Are there existing tools capable of performing similar tasks? 2) Assuming each observation has an abstract, could description/image retrieval be achieved by calculating text similarities [...]**
>
> This is a great question. The reason the target application is not possible with other methods (e.g., text similarity) is that not all Hubble images come with descriptive captions, like those used for fine-tuning the CLIP model (i.e., our observations are a small subset of all Hubble observations). Large survey-style campaigns image different parts of the sky without target-specific captions, which is the setting where the current method would be useful to query for phenomena of interest in large unlabeled datasets.
>
> **[...] is there some way to see how alinged the astronomical data with the CLIP model's domain? It is unclear whether the observed performance improvements are due to fine-tuning (from a generalist vision model) or if the original CLIP model had prior exposure to similar astronomical data, enabling rapid adaptation.**
>
> While this is difficult to check for the proprietary CLIP model used in our study, it should be possible to check for similarity of image embeddings in public datasets like LAION-5B. Unfortunately, these datasets are currently (May 2024) inaccessible due to safety concerns. However, given the domain-specialized nature of our data, it is highly unlikely that images of this kind were included in the CLIP trained set. The poor performance (Fig. 2 right) and lack of diversity of cosine similarities (Fig. 3 left) in the base model also indicate this. The fact that training from scratch does not match the performance of using the pre-trained model further indicates that a generalist backbone is necessary for performance.
>
> **Q:The paper extensively discusses the use of LLM-generated abstracts, though this approach does not yield superior results. Can the authors delve deeper into why this might be?**
>
> The [data-processing inequality](https://en.wikipedia.org/wiki/Data_processing_inequality) implies that the summarization process is necessarily lossy; the fact that the raw abstracts perform quite well implies that the CLIP training procedure is able to extract relevant information from the abstracts dataset, even though it is noisy. This capability was not a-priori expected, and is a central message of the study.

---

> > ### Comment · Reviewer_5XaM · 2024-06-04
> >
> > Text similarity baseline:
> > - I understood in practice there may not be summaries available. However, I think the test set you are using contains image summary pairs. I want a baseline here to better understand how hard the task is and how I interpreted these numbers.

---

> > > ### Author Response · Authors · 2024-06-07
> > >
> > > We thank the reviewer for their suggestion of further baselines. As also suggested by Reviewer dJXX, we performed a more quantitative evaluation of the query-based observation retrieval task, which we describe in the corresponding response.
> > >
> > > As suggested, we then also studied a comparison using the term frequency–inverse document frequency (tf-idf) on the proposal abstract, given the set of 10 benchmark queries. We found the following scores, for the average relevance (True or False) of the top 10 retrieved observations:
> > > **Base CLIP model:** 0.38
> > > **Fine-tuned CLIP model:** 0.77
> > > **tf-idf-based retrieval:** 0.82
> > >
> > > We note that the relatively high retrieval score of tf-idf is not very surprising, since the retrieved abstracts are likely to contain the queried terms and thus will be flagged as positively associated by the evaluator LLM. Nevertheless, it is a useful baseline to put the accuracy of CLIP models in context. We emphasize again that the main motivation for our method is to enable querying unlabeled datasets, where each image does not necessarily have a corresponding descriptive caption.

---

> > > > ### Comment · Reviewer_5XaM · 2024-06-07
> > > >
> > > > Thanks for the results. If we think of tf-idf-based retrieval as oracle, the result here actually shows that fine-tuned CLIP model is a very strong method.

---

### Official Review · Reviewer_dJXX · 2024-05-23

**Rating:** 5
**Confidence:** 4
**Ethics Flag:** 1

**Summary:**

The paper presents a new model called PAPERCLIP. The model is trained by fine-tuning CLIP on astronomical observations and abstracts from the Hubble telescope data. They train a variant of this model on summaries of the abstracts. Both these models show significantly higher performance than the base model on the held-out validation set. Further, the qualitative results on image retrieval are promising. Overall, this paper makes a unique contribution by adapting foundation models to astronomical data.

**Questions To Authors:**

Table 1 has two images: one from 1999 and the other from 2016. There are some obvious quality differences. Did you notice if these differences propagate to the model?

Long abstracts: The CLIP text encoder has a maximum text length of 77 tokens. What do you do when the abstract length is greater than 77 tokens? Do you discard it?

**Reasons To Accept:**

**Paper writing.** The paper is well written. All the sections are neatly organized, making it easy to understand the paper. The results and figures are very easy to follow.

**Unique contribution.** The paper presents a new dataset containing Hubble Space Telescope image observations and their abstracts. This dataset will be valuable to the community. The authors develop a specialized foundation model called PAPERCLIP by further fine-tuning CLIP on this dataset.

**Promising results.** The results of PAPERCLIP on the validation set show PAPERCLIP performs well whereas CLIP struggles with specialized datasets like the one introduced in the paper.

**Reasons To Reject:**

**Motivation.**

The need for PAPERCLIP needs to be better motivated in the paper. The authors say there is a “considerable interest in developing custom foundation models for the sciences”. While this may be true, it doesn’t say why creating a model like PAPERCLIP is necessary.
The qualitative evaluation focuses on image retrieval, but this setup is not well motivated. Are practitioners going to use the model to retrieve observations? If so, shouldn’t the quantitative experiments be designed to test image retrieval rather than showing accuracy on the validation set? (see evaluation weaknesses below).

**Technical contribution.**

PAPERCLIP which adapts a pretrained CLIP to the astronomical data with a contrastive loss objective. While this is a unique model, it is a classic example of transfer learning. Transfer learning is a well-studied area in machine learning. There is no new insight here.

Another contribution of the paper is summarizing the abstracts. The authors use the Outlines package to summarize the abstracts by adding constraints. But, the results in Figure 2 suggest that there is no advantage to using LLM-guided summaries. In Figure 3, the improvements are rather marginal. This suggests that the summaries may not be doing much to improve the model performance.


**Experiment and Evaluation.**

The quantitative evaluation is limited to one dataset. There is no evidence that this model works beyond this one dataset. Will the model perform well on other datasets not included in the training?

Evaluating the model’s performance on the validation set is not convincing. The performance curve only shows that loss drops on the validation set drawn from the same distribution. Furthermore, for completeness, the models trained with abstract should also be tested on the validation set containing the summaries.

The qualitative evaluation looks useful as one might query for images. However, the evaluation is limited to two objects. It would be great if the authors extended this setting to a formal benchmark for quantitative evaluation.

---

> ### Author Rebuttal · Authors · 2024-05-29
>
> **The need for PAPERCLIP needs to be better motivated in the paper.**
>
> Traditionally, astronomers train specialized models to search for phenomena of interest. This requires the curation of training data and separate models for each application. The motivation for this work is to introduce a method to train a single model that can be used to query relevant images from a set of observations, based on natural language.
>
> **Shouldn’t the quantitative experiments be designed to test image retrieval [...]?**
> **[...] one might query for images [...] It would be great if the authors extended this setting to a formal benchmark for quantitative evaluation.**
>
> The retrieval accuracy, as defined in Sec. 3.3, is precisely the image retrieval accuracy for an image-text pair validation set. Note that this metric is symmetric in text to/from image retrieval, when averaged over the entire validation set.
>
> **[...] the models trained with abstract should also be tested on the validation set containing the summaries.**
>
> We do exactly this in Fig. 3 (right) – here, all models (including trained with abstracts) are tested on the summaries, for a direct comparison.
>
> **Will the model perform well on other datasets not included in the training?**
> **Table 1 has two images: one from 1999 and the other from 2016. There are some obvious quality differences. Did you notice if these differences propagate to the model?**
>
> The dataset includes diverse images from Hubble, with different filters, targets, and post-processings. The differences between the images in Tab. 1 is coming from these variations, rather than differences in observation years. Our method is intended to be applied to a diversity of observations from a specific telescope, and we do not expect it to immediately generalize to different telescopes without further fine-tuning.
>
> **The CLIP text encoder has a maximum text length of 77 tokens. What do you do when the abstract length is greater than 77 tokens? Do you discard it?**
>
> We note in Sec. 3.2: “When using raw proposal abstracts, random chunks of the text delimited by periods are selected on the fly to fit within the maximum token length of the text encoder.” This also serves as a form of text data augmentation, given the relatively small dataset.

---

> > ### Comment · Reviewer_dJXX · 2024-06-04
> > **Reply to the Authors**
> >
> > Thank you for the clarification. Please highlight the motivation in the introduction. I am going to increase my score to 5.
> >
> >
> > Below I've shared some of my other concerns:
> >
> > My main concern is that the experiments do not necessarily support how the model will be used in practice. Figure 1 is misleading because it suggests the downstream task is "observation retrieval" with phrases. However, there is no quantitative evaluation for this experiment. The qualitative evaluation is not sufficient.
> >
> > In Section 4.2, you have the following sentence:
> > ```
> > We emphasize that for scientific usefulness, the goal is not necessarily to
> > correctly retrieve the most relevant objects, but rather to identify a diverse set of interesting
> > candidates for manual follow-up and further analysis.
> > ```
> > Even in the qualitative evaluation, the success criteria are not well defined. What does it mean for the candidate to be "interesting"? This is pretty vague.

---

> > ### Author Response · Authors · 2024-06-07
> >
> > We thank the reviewer for considering our response and raising the score. As suggested, we will highlight the motivation in the Introduction when revising the paper.
> >
> > **Quantitative evaluation of query-based retrieval task**
> >
> > To address the further concern of only showing qualitative evaluation for the “observation retrieval” task, we performed a quantitative evaluation of the relevance of retrieved images using the CLIP model. We designed the following prompt, used alongside Outlines to constrain the generated output to be boolean. This lets us evaluate whether the abstract corresponding to a retrieved observation is relevant or not.
> >
> > ```py
> > @outlines.prompt
> > def prompt_fn(abstract, query):
> >      """[INST]
> > You are an expert astrophysicist, with broad expertise across observational and theoretical astrophysics.
> >
> > Abstract: "{{abstract}}"
> > Query: "{{query}}"
> >
> > The above is an abstract for a proposed observation taken by the Hubble Space Telescope (labeled "Abstract"), and an object or concept (labeled "Query").
> >
> > Could the observations corresponding to the abstract contain the query? Be precise, and do not contain related concepts or objects.
> >
> > Your response should be either True or False. Only return True if the query is closely related to the abstract, and the downstream observation could be relevant to the query.
> > [/INST]
> > """
> > ```
> >
> > We then evaluated this prompt, for the base as well as fine-tuned models, on the top 10 closest images by cosine similarity returned for 10 different queries: ["globular cluster", "dwarf galaxy", "SN1987A", "strong lensing", "galaxy clusters", "interstellar medium", "dark matter", "spiral galaxies", "lyman alpha", "comets"]. We used the Mixtral-0.1-Instruct open-weights model utilized elsewhere in the paper.
> >
> > We obtained the following results for the average fraction of relevant images:
> > - **Base CLIP model:** 0.38
> > - **Fine-tuned CLIP model:** 0.77
> >
> > The fine-tuned model is thus significantly more likely to return images relevant to the query. We note that this test is still heuristic – as noted in the paper, the captions are noisy, and thus a “relevant” observation could be retrieved with a generic abstract (an example being Proposal 11956 in Tab. 4 of the paper). Nevertheless, this is a useful result to include in the revised paper, and we thank the reviewer for the suggestion.
> >
> > **What does it mean for the candidate to be "interesting"?**
> >
> > By "interesting", we mean that the retrieved images should be meaningfully related to the query, with the goal of guiding further inquiry. This is similar to e.g. searches in the space of materials or biomolecules, where relevant candidates can be identified with AI models and flagged for further domain-informed investigation. Figure 3 (left) provides quantitative evidence for this -- the fine-tuned model shows a clear separation in the distribution of cosine similarities between truly associated image-text pairs (solid red line) and randomly shuffled pairs (dashed red line), unlike the base model, highlighting its ability to surface relevant observations.

---

> > > ### Comment · Reviewer_dJXX · 2024-06-07
> > > **Reply to the Authors**
> > >
> > > Thank you for the additional results. They show that the fine-tuned CLIP model is better than the base CLIP. This is a positive result.
> > >
> > > I do have some more clarification questions:
> > > 1. The experiment setup is still not super clear to me. What is the input and the output? How does it relate to the downstream task in Figure 1? From what I can see in Figure 1, there is no need for Mixtral model or an abstract in the observation retrieval task.
> > > 2. What is this dataset? Is this a new dataset?
> > > 3. This is minor. From your reply, it seems that "a diverse set of interesting candidates" still means correctly retrieving relevant candidates. The sentence "We emphasize that for scientific usefulness, the goal is not necessarily to correctly retrieve the most relevant objects, but rather to identify a diverse set of interesting candidates for manual follow-up and further analysis." makes it sound like you are getting "diverse" objects rather than relevant objects.
> > >
> > > I agree with all the reviewers that this is a great extension of an existing technique to a new domain. I would be okay if the work was accepted to the conference.

---

> > > > ### Author Response · Authors · 2024-06-07
> > > >
> > > > We thank the reviewer for their positive assessment of the paper, and recommending acceptance to the conference. We would be grateful if an updated score could reflect this new assessment.
> > > >
> > > > 1. We are happy to further clarify the experiment, which closely follows the observation retrieval task in Fig. 1. We embed each of the 10 queries (these are the inputs) mentioned in the original reply using the text encoder, and retrieve the 10 most similar observations from the validation set by cosine embedding. We then prompt Mixtral by asking whether the associated abstract is relevant to the input query or not (with boolean output 1 or 0), and the quoted result is an average of this number over the 10 queries. The reviewer is correct that an external LLM is not needed for the retrieval task; however, in this case, it is used to evaluate the relevance of the retrieved observation by treating the (in this case available) associated abstract as the “ground truth”. This is done since there is no objectively correct evaluation in this case. This is closely related to LLM-based evaluations performed in some NLP benchmark tasks, when there isn’t an objectively correct answer.
> > > > 2. The dataset in this case is the same (validation) dataset used in the paper.
> > > > 3. We agree with the reviewer that this could be made clearer in the paper. By “diverse” candidates, here we simply mean that there may be many relevant observations retrieved, each interesting in its own way, without a fixed criterion. For example, when querying for a scientific use case (e.g., interstellar chemistry), observations containing many different types of objects (e.g., the interstellar medium, supernovae) could be retrieved, each being interesting and relevant. We will clarify this further in the revised paper.

---

### Decision · Program_Chairs · 2024-07-10

**Decision:**

Accept

**Comment:**

The reviewers are generally in agreement that the paper’s primary contribution is that it applies LLMs (or, more precisely, a VLM) to the interesting and novel domain of astronomical data. The concerns raised are primarily that 1) the method is not overwhelmingly novel (i.e., it is just finetuning CLIP) and 2) there are some questionable methodological choices, specifically a lack of good baselines or designated test sets.

There was a lively discussion between authors and reviewers, which at least partially addressed the above concerns by providing some additional experiments which compared CLIP to the fine-tuned CLIP and to tf-idf on a retrieval task, and which justified the experimental design choices as being the result of data limitations.

Taking everything into account, I believe this paper is worth including in COLM. It is primarily an engineering paper, but it represents the type of important engineering work that can be difficult to publish in traditional single-discipline conferences, and which will be instrumental in if/how LLMs impact the (scientific) world more broadly. This, as I understand it, is in line with the mission of COLM. I think it is valuable to highlight proofs-of-concept for adapting LLMs/VLMs to new domains, and that this can serve as an exemplar and/or a starting point for similar efforts in astronomy, and in other domain-specific LLM adaptations.

The reviewers provided very valuable critiques and suggestions during the discussion. I appreciate the authors’ good responses during discussion, and the authors should consider it necessary that the paper is updated to reflect the outcomes of this discussion prior to publishing.

[At least one review was discounted during the decision process due to quality]